# Sorafenib Nanomicelles Effectively Shrink Tumors by Vaginal Administration for Preoperative Chemotherapy of Cervical Cancer

**DOI:** 10.3390/nano11123271

**Published:** 2021-12-01

**Authors:** Jun Wang, Fengmei Lv, Tao Sun, Shoujin Zhao, Haini Chen, Yu Liu, Zhepeng Liu

**Affiliations:** 1Department of Pharmaceutics, School of Pharmacy, Fudan University & Key Laboratory of Smart Drug Delivery (Fudan University), Ministry of Education, Shanghai 201203, China; wangjun245186@126.com; 2School of Medical Instrument and Food Engineering, University of Shanghai for Science and Technology, Shanghai 200093, China; LFMusst@126.com (F.L.); suntao930812@163.com (T.S.); shoujinzhao@163.com (S.Z.); HN_C383@163.com (H.C.)

**Keywords:** sorafenib, micelles, cervical cancer, vaginal administration, preoperative chemotherapy

## Abstract

To investigate the potential of sorafenib (SF) in preoperative chemotherapy for cervical cancer to reduce tumor volume, sorafenib micelles (SF micelles) with good stability and high drug loading were designed. SF micelles were prepared by film hydration followed by the ultrasonic method. The results showed that the SF micelles were spherical with an average particle size of 67.18 ± 0.66 nm (PDI 0.17 ± 0.01), a considerable drug loading of 15.9 ± 0.46% (*w/w*%) and satisfactory stability in buffers containing plasma or not for at least 2 days. In vitro release showed that SF was gradually released from SF micelles and almost completely released on the third day. The results of in vitro cellular intake, cytotoxicity and proliferation of cervical cancer cell TC-1 showed that SF micelles were superior to sorafenib (Free SF). For intravaginal administration, SF micelles were dispersed in HPMC (SF micelles/HPMC), showed good viscosity sustained-release profiles in vitro and exhibited extended residence in intravaginal in vivo. Compared with SF micelles dispersed in N.S. (SF micelles/N.S.), SF micelles/HPMC significantly reduced tumor size with a tumor weight inhibition rate of 73%. The results suggested that SF micelles had good potential for preoperative tumor shrinkage and improving the quality life of patients.

## 1. Introduction

Cervical cancer is a global health problem [1]. Even in developed countries, due to population migration, incomplete screening and limitations of vaccination programs, cervical cancer is still a common gynecological malignant tumor with a high degree of malignancy. Surgery is the main treatment for early-stage cervical cancer, but for patients with intermediate and advanced cervical cancer, the opportunity for surgery is often lost. Therefore, it has now been developed to apply chemotherapeutic drugs before surgery to reduce the size of tumor and improve tumor resection rate [2], to ensure less damage to pelvic tissues, to maintain maximum vaginal length and elasticity and to improve the quality of patient’s life under radical hysterectomy [3,4,5,6].

At present, cytotoxic treatment of cervical cancer is mainly based on cisplatin combined with chemotherapy. While killing tumor cells, it also kills a large number of normal cells, resulting in a series of toxic side effects [7]. Drugs with targeting properties that improve efficacy while reducing toxic side effects have been a hot spot in recent years for research into the treatment of cervical cancer at home and abroad. VEGF (vascular endothelial growth factor) can directly promote tumor vascular endothelial cell differentiation and proliferation, which is closely related to tumor growth, infiltration and metastasis. Research shows that cervical cancer tissues highly express VEGF [8]. Sorafenib (SF) is a small molecular and multi-target drug. It can inhibit the proliferation of tumor cells by blocking RAF/MEK/ERK-mediated cell signal pathway, and also inhibit tyrosine kinase receptors such as VEGF-2 and VEGFR-3 [9]. So, in the current phase I/II clinical trials, tyrosine kinase inhibitors as sorafenib, have been used to inhibit tumor growth and angiogenesis by inhibiting the VEGF signaling pathway [10,11]. However, SF is insoluble in water and has difficulties in delivery. It is clinically prepared as an oral formulation, which limits its application for low bioavailability [12,13] and high dosage [14]. Therefore, it is necessary to develop a new formulation of SF to achieve the ultimate goal of improving water solubility and distribution within tumors.

Researchers have explored many nano-drug deliveries of SF, such as micelles [15,16,17], liposomes [18,19,20], polymer nanoparticles [21] and inorganic nanoparticles [22]. However, most of these nanomaterials are not approved by the FDA. D-alpha-Tocopheryl polyethylene glycol 1000 succinate (TPGS) is an FDA certified excipient [23,24,25]. It is a safe, excellent and amphiphilic nonionic surfactant, which can increase the solubility and the stability of micelles [26]. It can also improve the membrane permeability by inhibiting the efflux of p-gp, increase accumulation and then improve anti-tumor efficacy [27].

Therefore, in this study, SF micelles were prepared using TPGS, which significantly improved the water solubility of SF. HPMC (hydroxypropyl methylcellulose) is a bioadhesive polymer that spreads easily and adheres to the intravaginal to prevent rapid drug loss, by dispersing the SF micelles in the bioadhesive HPMC, prolonged the residence of SF micelles in vagina [28], thereby increasing the distribution of SF in the tumor and improving the therapeutic effect of cervical cancer (Figure 1).

## 2. Materials and Methods

### 2.1. Materials

SF, TPGS, hydroxypropyl methylcellulose (HPMC, type I, viscosity of 2% (*w*/*w*) aqueous solution at 20 °C: 4000 mPa s), 2-(4-amidinophenyl)-6-indolecarbamindine dihydrochloride (DAPI), 6-coumarin (C6), LysoTracker Red DND-99, DiR iodide, Thiazolyl Blue Tetrazolium Bromide (MTT) were purchased from Meilun Biotechnology Co., Ltd. (Dalian, China). All other chemicals were of analytical grade, purchased from Sinopharm Reagent Co., Ltd. (Shanghai, China) and used as received.

TC-1, a cervical cancer-related cell line was purchased from Heyuan Biotechnology Co. Ltd. (Shanghai, China), was cultured in RPMI-1640 complete medium containing 10% FBS (Gibco, ThermoFisher Co., Waltham, MA, USA), supplemented with 100 U/mL penicillin and 100 μg/mL streptomycin to prevent cell infection. The humidity of the culture environment was 95%, the temperature was 37 °C, and the CO_2_ concentration was 5%. Cellular culture dishes, confocal dishes and cell culture plates were purchased from Cellvis (Cellvis, Mountain View, CA, USA).

Female C57BL/6 mice (6–8 weeks) were from the Shanghai Laboratory Animal Center (Shanghai, China, production license number SCXK Shanghai 2017-0005) and kept under SPF conditions. Animal experiments were conducted under the guideline approved by the Ethics Committee of the School of Pharmacy, Fudan University (2021-03-SL-LZP-31).

### 2.2. Preparation of Micelles

Dissolve 40 mg TPGS and different masses of SF (1, 1.5, 2, 4, 8, 10, 15 and 20 mg) in 2 mL ethanol, and the mixture was mixed well using a vortex, followed by transfer to a 50 mL eggplant-shaped flask and evaporated in a rotary evaporator (IKA, Staufen, Germany) at defined temperatures (37 °C) using 100 rpm to form a thin film. The rotary evaporation time was about 10 min, and the angle of inclination of the eggplant bottle was adjusted to maximize the contact area between the liquid and the inside of the flask so that the film formed was thin, transparent and homogeneous. Immediately after film formation, 5 mL of normal saline (N.S., 0.9%, *w*/*v*) was added and gently mixed at 25 °C for hydration (the hydration process shall be rapid in order to avoid moisture absorption of the dry film when exposed to air, which would affect the hydration effect), in the meantime the nitrogen blowing needle was dipped into the liquid and the nitrogen flow rate was adjusted to form tiny bubbles during hydration. Finally, the obtained suspension was sonicated in an ice bath for 5 min (100 W) to get SF micelles. The method for preparing fluorescently labeled micelles was similar, but in the first step, an additional 25 μg C6 or 50 μg DiR was added to the 2 mL ethanol in the eggplant flask.

### 2.3. Preparation of Micelles Loaded HPMC

To increase the sustainability of the micelles in contact with the mucosa and to release the drug slowly, SF micelles dispersed in HPMC (SF micelles/HPMC) were obtained by dissolving an appropriate amount of HPMC (5%, *w*/*w*) in SF micelles at 25 °C. The specific operation steps were to first weigh 5 mL of SF, then calculate the mass of HPMC so that HPMC accounts for 5% of the total mass, and then add HPMC to SF micelles in small amounts several times, stirring (200 rpm) while adding until a uniform translucent gel was formed. DiR-micelles/HPMC were also prepared similarly.

### 2.4. Characterization

#### 2.4.1. Particle Size and Distribution

The micelles were diluted with ultrapure water to a SF concentration of 100 μg/mL and then the particle size and zeta potential of micelles were measured at 37 °C applying the Zetasizer instrument system (ZS-10-82, Malvern Instruments Co., Ltd., Malvern, UK) through dynamic light scattering (DLS) technology.

#### 2.4.2. Morphology

The morphology of SF micelles was observed on Tecnai G2F20S-TWIN (FEI Co., Hillsboro, OA, USA) by transmission electron microscopy (TEM). Briefly, 20 μL SF micelles with SF concentration of 60 μg/mL was dropped on the copper grid with carbon film attached on the surface. After 15 min, the excess sample was absorbed by the burrs of the filter paper so that there was only a thin layer of the sample on the copper grid without droplets. Then the copper lattice was naturally dried at room temperature and observed by TEM.

#### 2.4.3. Drug Loading (DL) and Encapsulation Efficiency (EE)

Measurement of UV detection wavelength of SF micelles: SF and TPGS solutions were prepared with acetonitrile at 100 μg/mL, respectively, and SF micelles were diluted with acetonitrile until the SF was 50 μg/mL (so that their absorbance values were between 0.2 and 1), and 800 μL was placed in a quartz cuvette and the absorption spectra were scanned at 200~400 nm by a UV spectrophotometer to determine the maximum absorption wavelength.

The freeze-dried SF micelles were accurately weighed and dissolved in an appropriate volume of acetonitrile to make the concentration within the linear range. After filtered with a 0.22 μm syringe filter, the filtrate was analyzed by an Agilent 1100 HPLC system (Palo Alto, CA, USA). The HPLC condition was: Diamonsil^®^ C18 column (250 mm × 4.6 mm, 5 μm, Dikma Technology, Beijing, China), MeOH:KH_2_PO_4_ (pH = 2.5, 70:30, *v:v*), 0.7 mL/min, 262 nm. DL and EE were calculated using the following equations.
(1)$$\mathrm{EE}\;(\%)=\frac{\mathrm{Amount}\;\mathrm{of}\;\mathrm{drug}\;\mathrm{in}\;\mathrm{micelles}}{\mathrm{Total}\;\mathrm{amount}\;\mathrm{of}\;\mathrm{feeding}\;\mathrm{drug}}\times \;100$$
(2)$$\mathrm{DL}\;(\%)=\frac{\mathrm{Amount}\;\mathrm{of}\;\mathrm{drug}\;\mathrm{in}\;\mathrm{micelles}}{\mathrm{Total}\;\mathrm{amount}\;\mathrm{of}\;\mathrm{micelles}}\times \;100$$

#### 2.4.4. Crystalline Characterization

The SF micelle lyophilized powder was analyzed by X-ray powder diffraction (XRPD) using a D2 Phaser diffractometer (BrukerCorp, Billerica, MA, USA). Test conditions: The voltage was 40 kV. The current was 100 mA. The rotation range of the diffraction angle (2θ) was 0 to 50°, the scanning speed was 5 °/min, and the step size was 0.02°.

Differential scanning calorimeter (DSC) analysis of micelles and bulk drugs was measured by a Perkin–Elmer Pyris 1 DSC instrument (Waltham, MA, USA) with an Intra-cooler 2P cooling accessory. Preweighted lyophilized powders were sealed into aluminum crucible (NETZSCH, Selb, Germany) and place an empty aluminum crucible as a reference. The nitrogen flow rate was 10 mL/min, the heating rate was 10 °C/min, and the scanned temperature ranged from 20 to 300 °C.

#### 2.4.5. Stability

The time-dependent stability of SF micelles in different media including PBS, RPMI 1640 medium containing 10% FBS, and FBS were evaluated for 48 h (concentration of SF equivalent to 300 μg/mL). At each time point, 200 μL micelles in different media were taken out and the particle size was measured using the DLS, while another 200 μL was removed to monitor aggregation by detect their absorbance at 560 nm (560 nm as the turbidity characteristic wavelength) using the Synergy 2 (Biotek, Green Mountains, VT, USA).

#### 2.4.6. In Vitro Release

In vitro release behavior of SF micelles were detected by bag filter method with vaginal fluid simulant (VFS) [29] containing 1% Tween 80 (*w*/*w*) as the release medium. Briefly, 1 mL SF micelles or SF bulk (SF concentration 500 μg/mL) were added into SnakeSkin™ dialysis bags with MWCO 35000Da (ThermoFisher, Waltham, MA, USA), and then completely immersed into 10 mL release medium and continuously shook under 100 rpm by a reciprocating shaker bath (THZ-103B, Shanghai Hengyi Scientific Instruments Co. Ltd., Shanghai, China) at 37 °C. 400 μL samples of different groups were taken from the medium at predetermined time intervals and replaced with an equal volume of fresh medium, and the released SF concentration was analyzed by HPLC to obtain the cumulative release percentage. HPLC conditions were similar to Section 2.4.3.

### 2.5. Cellular Experiments

#### 2.5.1. Cellular Uptake

The uptake of SF micelles by cervical cancer cells was carried on TC-1 cells, which exhibited similar genetic traits to cervical tumors induced by human papillomavirus (HPV) [30]. For fluorescent imaging, TC-1 cells were inoculated onto the 12-hole plate at the density of 2 × 10^5^ cells per hole. After 12 h incubation, the medium was replaced with fresh medium containing C6-labeled SF micelles and Free C6 (the terminal concentration of C6 was 10 nM). After 4 h incubation, the medium was removed and the cells were washed twice with PBS to avoid the interference of unbound SF micelles. We treated the cells with 4% paraformaldehyde fix solution for 15 min, stained the nulei with DAPI for 3 min and finally took fluorescence images with a fluorescence microscope (Leica, DMI4000 B, Wetzlar, Germany).

For colocalization with lysosomes, cells which had been incubated with C6-labeled SF micelles and Free C6 were treated with 37 °C preheated LysoTracker Red DND-99 (50 nM) for 45 min and then fixed with paraformaldehyde fix solution for 15 min, finally treated with DAPI for 3 min to track its position. Fluorescent images were captured by Leica confocal microscope (Leica, TCS SP8, Germany). The excitation (Ex) and emission (Em) parameters were: DAPI Ex 364 nm/Em 461 nm, C6 Ex 466 nm/Em 504 nm and LysoTracker Red Ex 577 nm/Em 590 nm. The colocalization coefficient of fluorescence was analyzed by ImageJ software (Image J 2x 2.1.4.7 Wayne Rashand, National Institutes of Health, Bethesda, MD, USA).

For flow cytometry, the process was similar to the above cell uptake steps. The difference was that the cells were digested with trypsin and then collected by centrifugation (1000 rpm, 10 min) after being washed twice with PBS. The collected C6 labeled cells were finally resuspended in PBS and counted by a flow cytometer BD FACSCalibur Flow Cytometry System (Applied Cytometry Systems, Franklin Lakes, NJ, USA).

#### 2.5.2. In Vitro Cytotoxicity

The TC-1 cells were seeded into a 96-well plate at a seeding density of 3 × 10^3^ cells per well. After 12 h incubation, the cells were treated with fresh medium, to which SF micelles and Free SF were added to provide the final concentration of SF equivalent to 12.91 μM, 17.21 μM, 20.65 μM, 32.27 μM, 43.03 μM, 51.63 μM and 64.54 μM, respectively. After 24 h incubation, cell viability was measured by MTT method. Briefly, TC-1 cells were treated with MTT (5 mg/mL, 20 μL per well) for 4 h. The medium was removed with a 5 mL syringe and the formazan crystal was fully dissolved in dimethyl sulfoxide (DMSO). Absorbance data was performed by Power Wave XS Microplate Spectrophotometer at 490 nm (BioTek Instruments, Inc., Winooski, VT, USA).

Preparation of free sorafenib solution: since sorafenib is almost insoluble in water, we followed the formulation of Taxol^®^ in this experiment [31], that is, a mixture of polyoxyethylene castor oil: ethanol = 1:1 (v:v) to dissolve sorafenib, and after sonication to obtain a free sorafenib solution (Free SF).

### 2.6. Characterization of HPMC as a Micellar Dispersion Carrier

HPMC was added to the SF micelles to a mass fraction of 5% and then took 1 mL and placed it directly under the rotor (this process should be done as slowly as possible to avoid bubbles in the sample, and extra sample could be scraped off using a flat scraper) and tested for elastic (G′) and viscous modulus (G″) by rheology (Bohlin Gemini 2, Malvern Panalytical, Malvern, UK). The parameters were as follows: Mode: oscilllation, Temperature: 25 °C, Frequency: 1 Hz, Measuring System: PP40 Acrylic, the correlation curves of G′ and G″ vs. strain were obtained at the end of the measurement.

### 2.7. In Vivo Intravaginal Residence

10 μL DiR-labelled micelles dispersed in HPMC (DiR-micelles/HPMC) or saline (DiR-micelles/N.S.) was intravaginally administered into healthy female ICR mice by operating a microliter syringe equipped with a smooth blunt needle as previously reported [32]. Fluorescence images were taken at 0, 1, 3 and 5 h after administration with a fluorescence in vivo imaging system (IVIS Spectrum, PerkinElmer, Santa Clara, CA, USA) at Ex 748 nm/Em 780 nm.

### 2.8. In Vivo Evaluation in TC-1-Luc Model Tumor-Bearing Mice

Six- to eight-week-old female C57BL/6 mice were purchased from Shanghai Slack Laboratory Animal Co., Ltd. (license number: production license number SCXK Shanghai 2017-0005) and housed in an SPF animal room. All animal experiment conforms to the ethical standards of animal experiment and was approved by the Experimental Animal Ethics Committee of the School of Pharmacy of Fudan University. The orthotopic cervical model of C57BL/6 mice were established referring to the method of Ci et al. [33]. Mice were subcutaneously injection with β-estradiol (0.1 mg/day) for a successive seven days before tumor inoculation. Then mice were anesthetized by intraperitoneal injection of 8% chloral hydrate (*w*/*v*, 100 μL per mouse) and placed them in a supine position. Finally, sterilized dry cotton balls were moistened with VFS to softly wipe off vaginal secretions, and a sterilized cytobrush was used to gently disrupt the cervicovaginal epithelium. TC-1-Luc cells were diluted to required concentration and then inoculated intravaginally at 1 × 10^5^ cells per mouse. To avoid a drop in body temperature during anesthesia, the mice were placed on an insulating pad to keep them warm until they awoke and then placed back in their cages. The successful establishment of the tumor model was confirmed by palpation, anatomical and histological examination. When it was found that TC-1 tumors volume grew to about 100 mm^3^, it was defined as day 0 and intravaginal administration was started. The mice were anesthetized at a low dose after administration and kept in a supine position for 20 min to avoid the mice licking. We employed intravaginal administration because local administration for cervical cancer could effectively release the drug at the lesion site and allow full play to its anti-tumor effect [34,35,36]. We used a 10 μL syringe equipped with a blunt needle for intravaginal administration, and the needle was dipped into the VFS before administration to obtain lubricity and to reduce mucosal damage as well as the pain to the mice when the needle entered and exited the vagina.

Mice were randomly divided into five treatment groups (n = 6/group) and intravaginally treated with N.S., HPMC, empty micelles, Free SF/HPMC (15 mg/kg) and SF micelles/HPMC (15 mg/kg). Since HPMC was insoluble in ethanol, Free SF/HPMC was prepared in in vivo anti-tumor experiments by first dissolving SF with DMSO, then diluting it with ultrapure water to the required concentration for in vivo administration, and finally adding HPMC in small amounts several times to obtain Free SF/HPMC. The drug was administered intravaginally every 3 days, and the body weight and tumor bioluminescence intensity were monitored. D-Luciferin at 150 mg/kg was injected intraperitoneal into the tumor of TC-1-Luc cervical cancer model 10 min prior to whole-body imaging. Use the Xenogen IVIS^®^ 200 Series to obtain whole-body bioluminescence images, and the region of interest (ROI) was acquisition and quantification applying Living Image^®^ software. On day 16, mice were sacrificed to resecting the tumors connecting uterus and uterine tubes for photographing (to prove that they were orthotopic tumors), weighing the tumors and further H&E (Hematoxylin-eosin) microscopic examination of tumors, vaginas and organs. The relative bioluminescence intensity was calculated as Formula (3) and the inhibition rate of tumor weight was calculated as Formula (4):(3)$$\mathrm{Relative}\;\mathrm{bioluminescence}\;\mathrm{intensity}=\frac{V_{\mathrm{bioluminescence}\;\mathrm{intensity}\;\mathrm{on}\;\mathrm{the}\;\mathrm{day}\;\mathrm{of}\;\mathrm{measurement}}}{V_{\mathrm{Bioluminescence}\;\mathrm{fluorescence}\;\mathrm{intensity}\;\mathrm{on}\;\mathrm{day}\;0\;}}\times 100\%\;$$
(4)$$\mathrm{Inhibition}\;\mathrm{rate}\;\mathrm{of}\;\mathrm{tumor}\;\mathrm{weight}=\frac{W_{\mathrm{saline}\;\mathrm{group}}{-W}_{\mathrm{treatment}\;\mathrm{group}}}{W_{\mathrm{saline}\;\mathrm{group}}}\times 100\%\;$$

### 2.9. Statistical Analysis

All statistical analysis was performed using GraphPad Prism 8 (La Jolla, CA, USA). The statistical differences among groups were conducted by the two-way ANOVA test. A value of *p* < 0.05 was considered significant and *p* < 0.01 was considered highly significant.

## 3. Results and Discussion

### 3.1. Optimization

SF micelles were prepared by thin-film hydration method [37], and SF micelles with different ratios of SF:TPGS were placed in small glass bottles to observe the Tyndall effect. The beam passed through all different ratios (Figure A1), indicating that the fresh SF micelles were uniformly dispersed in water. The best ratio of SF to TPGS was further optimized by particle size, polydispersity (PDI), zeta-potential, drug loading (DL) and entrapment efficiency (EE). The results were shown in Table A1, with the increase of the SF from 1:40 to 8:40, the drug loading of SF in micelles increased from 1.91% to 15.9%, and the average particle size increased from less than 40 nm to 67 nm (Figure A2), which was due to the mass of TPGS remained constant and as the SF increased, more SF was loaded into the core of the micelles, thus increasing the particle size and drug loading. The encapsulation efficiency was decreased but not significantly different, all being greater than 89%. It was worth mentioning that with the increased of SF from 1:40 to 4:40, the particle size did not gradually increase which might be due to the excessive TPGS forming small empty micelles, which interfere with the particle size results. While continue to increase the SF content to a ratio higher than 8:40, the encapsulation efficiency dropped sharply, probably because the drug has exceeded the carrier TPGS encapsulation. Whereas the drug loading still increased probably possibly due to some of the SF formed nanocrystals with a small amount of TPGS, which was a thermally unstable system with high drug loading [38], hence the drug loading increased. However, due to nanocrystal instability, 10:40, 15:40 and 20:40 all precipitated after being stored at room temperature for 2 h, therefore SF:TPGS = 8:40 (1:5) was selected for the following experiment.

### 3.2. Successful Preparation of SF Micelles

SF micelles with optimized ratio 1:5 (SF:TPGS) were successfully prepared as shown in Figure 1. The SF micelles exhibited a near-spheroid shape with a size of 67.18 ± 0.66 nm (Figure 1A). The zeta potential of SF micelles was −1.54 ± 0.8 mV (Figure 1B). UV spectrum result displayed that SF and TPGS had the maximum ultraviolet absorption wavelengths at 205 nm and 262 nm (Figure A3), respectively, SF micelles had absorption wavelengths for both, since the large difference between 205 nm and 262 nm, the presence of TPGS would not interfere SF measurements. The DL and EE of SF micelles measured by HPLC at 262 nm were 15.9 ± 0.46% and 89.4 ± 0.19%, respectively (Table A1).

In the XRPD spectrum (Figure 1D), bulk SF displayed diffraction peaks at 10.7°, 19.1°, 22.8°, 23.3° and 25°, bulk TPGS displayed diffraction peaks at 8.5°, 9.0°, 11.0°, 17.0° and 17.7°, the physical mixture of bulk SF and bulk TPGS showed both two separate characteristic peaks, while SF micelles showed diffraction peaks at 29.10° and 33.50°, indicating that the crystallinity structure of SF micelles was different from that of bulk SF, and SF micelles were not a simple physical mixture of SF and TPGS. DSC further verified their difference in crystal structures (Figure 1E), bulk SF and bulk TPGS displayed an exothermic peak at about 217 °C and 107 °C, respectively, and the physical mixture of bulk SF and bulk TPGS remained the endothermic peak of bulk SF, while there was no such characteristic peak in SF micelles, but displayed an endothermic peak around 60 °C, which suggested that mixing SF and TPGS might not change the crystalline of SF; however, the formulation SF into SF micelles, the crystalline was changed.

From Figure 2A, the stability of SF micelles was evaluated in different mediums including 0.01 M PBS, RPM1640 + 10% FBS, and FBS, respectively. The particle size results measured by DLS showed that micelles could be stable for at least 48 h in all three media, while the particle size measured in RPM1640 + 10% FBS and FBS media was larger than that measured in PBS, which might be due to the protein crown formed by FBS on the particle surface, thus we further used the previously reported method to assess the agglomeration of particles in the presence of FBS by measuring the absorbance of micelles in different media at 560 nm [39,40,41,42]. The results indicated that the SF micelles remained good stability in both PBS and serum-containing PBS buffer (Figure 2C).

Compared with Free SF, SF micelles had a slow-release effect and could be completely released within 72 h (Figure 2D), providing evidence for the dosing interval in subsequent animal experiments (every 3 days). Compared with the release properties of SF micelles in saline (SF micelles/N.S.), the sustained release effect of SF micelles loaded with HPMC (SF micelles/HPMC) was more obvious. It took about 12 h for SF micelles/N.S. while about 24 h for SF micelles/HPMC to reach a cumulative release of 50%. We observed that the cumulative release of Free SF/HPMC reached a plateau at about 24 h and was less than 50% at 72 h, which might be due to the precipitation of Free SF in HPMC.

### 3.3. In Vitro Cellular Uptake and Cytotoxicity of Micelles

For fluorescent images, SF didn’t have fluorescent properties and could not track its uptake behavior in cells. Therefore, a general fat-soluble fluorescent C6 (Free C6) was selected to represent free sorafenib [43,44]. As shown in Figure 3A–C, cellular uptake of Free C6 was less than that of SF micelles. The intracellular fluorescence in cells was stronger after incubated with C6-labeled SF micelles and most of which located around the nucleus and overlapped with lysosomal fluorescence, while the fluorescence of Free C6 was much weaker and seemed scattered in the cytoplasm, and only a small part of it overlapped with the lysotracker. (Figure 3D–F).

Cytotoxicity was evaluated at different concentrations of SF in SF micelles and Free SF. After 24 h incubation, IC_50_ of Free SF and SF micelles was 76.65 μM and 46.76 μM, respectively. SF micelles exhibited a significant antiproliferative effect on tumor cells at low concentrations. It was worth mentioning that empty micelles had no obvious cytotoxicity at all concentrations, which proved the safety of TPGS (Figure 4A). Flow cytometry (Figure 4B) revealed higher cellular uptake of SF micelles than that of Free C6 after 4 h co-incubation as well. The apoptosis results showed that cells treated with empty micelles for 24 h had no obvious apoptosis, demonstrating the good biocompatibility of TPGS, while both Free SF and SF micelles had apoptotic effects on the cells with apoptosis rates of 10.12 ± 0.49% and 23.76 ± 1.37%, respectively (Figure 4C,D). It indicated that SF micelles could induce apoptosis in TC-1 cells to some extent.

### 3.4. SF Micelles Dispersed in HPMC Facilitate Intravaginal Retention

For in vivo experiences, SF micelles were dispersed in HPMC to avoid immediate leakage caused by administered as an aqueous dispersion. The adhesion of HPMC enhanced the contact between micelles and mucosa, so that the SF had time to be absorbed and achieve the purpose of treatment. HPMC was a practical vaginal administration vehicle with the advantages of safety, simple preparation process and good drug compatibility [45]. A 5% (*w*/*w*) concentration of HPMC was selected to deliver SF micelles. The elastic (G′) and viscous moduli (G″) confirm the adhesion of HPMC. As shown in Figure 5, the G′ and G″ of SF micelles dispersed in N.S. (SF micelles/N.S.) were less than 1 Pa, while the G′ and G″ of SF micelles dispersed in HPMC (SF micelles/HPMC) increased to more than 1000 pa. We also monitored the particle size variation of SF micelles loaded into HPMC. Before measuring the particle size of SF micelles/HPMC with DLS, it was diluted with ultrapure water to obtain good fluidity and solution-like state, and then the particles were fully dispersed by bath ultrasonic for 30 s. The results showed an increase from 67 nm to about 178 nm (Figure A4), which was still conducive to penetrating mucus due to its nanoscale less than 500 nm [46,47].

Before intravaginal administrated DiR-micelles/HPMC and DiR-micelles/N.S., vaginals were wash with VFS to achieve the purpose of cleaning. Fluorescence photos and semi-quantitative results indicated that SF micelles/HPMC has a prolonged residence (Figure 6).

### 3.5. Tumor Shrinkage Effect in the Orthotopic TC-1 Cervical Cancer Model

It was reported that the orthotopic TC-1 model strictly represents the cervicovaginal cancer in a short time [48]. The bioluminescence images of tumor-bearing mice were shown in Figure A5 and Figure 7, and there was a near linear correlation between bioluminescence signal and tumor weight as reported [49,50,51], indicating that the bioluminescence TC-1 tumor model could accurately assess tumor size. On the 16th day after intravaginal administration, the bioluminescence intensity of the N.S., HPMC and Empty micelles/HPMC was approximately 155, 156 and 160 times higher than that at day 0 (Figure 7A,B), respectively, with no significant difference between the three groups. The bioluminescence intensity of the Free SF/HPMC group was about 120 times higher than that at day 0, while SF micelles/HPMC was about 76 times higher than that at day 0. The average tumor weight of SF micelles/HPMC was 1.27 g (Figure 7C), the inhibition rate of tumor weight was 72.77% (Figure A7), and the tumor photographs showed that SF micelles/HPMC had a better tumor shrinkage effect than the Free SF/HPMC (Figure A6). These results indicated that SF micelles/HPMC inhibited tumor growth to the highest extent. There was no significant difference in body weight for all groups (Figure 7D). No obvious difference was observed in H&E stained-section of major organs among all groups (Figure A8), H&E stained sections of the vagina in the Free SF group showed that the mucosal epithelium was destroyed, while the mucosal structure in the SF micelles group was intact (Figure 7E), indicating that SF micelles had no significant toxic side effects. All data indicated that SF micelles could safely and effectively inhibit tumor growth.

We noticed that before day 8, there was no significant difference in the bioluminescence intensity between Free SF and SF micelles, whereas after day 8, Free SF was less effective in inhibiting tumor growth compared with SF micelles because as TC-1 model tumor grew, elevated interstitial pressure within tumor impaired the penetration of free drugs, making it difficult for the Free SF to further reduce the tumor volume [52].

### 3.6. Discussion

In this study, SF micelles were formulated by the film hydration method [37,53], inspired by the “CO_2_-assisted method”, a method that using impetus of CO_2_ bubbles to reduce the particle size [54], we immersed the nitrogen blowing needle into the liquid during the hydration process, and used the N_2_ bubbles to achieve rapid micro-mixing, resulting in a uniform distribution of SF and TPGS in the hydration solution, thus facilitating the acquisition of SF micelles with homogenous particle size. After hydration, a probe ultrasonic process was adopted to increase the chance of interaction between the hydrophobic end of TPGS and the hydrophobic drug SF by using ultrasound input energy for strong mixing, and then to increase the drug loading.

The micelles with particle size less than 30 nm were formed by self-assembly of amphiphilic properties, while the particle size in this study was larger than 50 nm. Considering that SF micelles were prepared using probe ultrasound, it has been reported that ultrasound can break the micelles into smaller micelles fragments and use these fragments as nuclei to promote micelle growth and finally obtain homogeneous micelles [55]. Therefore, SF micelles may be formed by a “micelle aggregation” [56], which aggregated from small simple micelles to form large complex micelles.

The tumor shrinkage effect of Free SF/HPMC was far less than that of SF micelles/HPMC, which might be due to the hydrophobicity of the Free SF, which easily precipitated in the hydrophilic HPMC environment, and as rapid proliferation of tumor cells, which elevated the interstitial hydraulic pressure and prevented the precipitated particles from entering the dense tumor [51]. In contrast, the hydrophilic chains of TPGS on the surface of SF micelles maintained the stability of SF micelles in HPMC and reduced the adhesion of micelles to mucus, allowing SF micelles to penetrate the mucus to reach the tumor site [57].

## 4. Conclusions

In conclusion, by using the amphiphilic polymer TPGS as a carrier, we successfully prepared SF micelles that improved the solubility of SF, and showed higher cell uptake and stronger cell growth inhibition compared with Free SF. SF micelles/HPMC exhibited prolonged intravaginal retention in vivo, thus improving the anti-tumor efficacy. Therefore, as a preoperative local chemotherapeutic formulation for cervical cancer, SF micelles/HPMC had a good potential for preoperative tumor shrinkage, which was expected to enhance the surgical resection rate of the lesion and improve the postoperative quality of life of the patients.

## Data Availability

Not applicable.

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
