# Peer review of "Sorafenib Nanomicelles Effectively Shrink Tumors by Vaginal Administration for Preoperative Chemotherapy of Cervical Cancer"

_nanomaterials, 2021, doi:10.3390/nano11123271_

Round 1
Reviewer 1 Report
The article refers to the encapsulation of sorafenib (SF) in micelles (using different surfactants) and its potential to be used in preoperative tumor shrinkage. Different micelles were prepared and characterized and the in vitro release of the drug evaluated, as well as the cellular intake and cytotoxicity. Finally, in vivo experiments shows a high potential of the system.
Although the topic is of great interest, the manuscript lacks some important aspects and for this reason, I recommend its publication after mayor revisions.
Regarding the synthesis and characterization of the micelles:
- The Materials and Method section of the manuscript, which is expected to be the most well described part of the work, lacks of important details that makes the preparations of the micelles non-reproducible. In this sense, more details should be included, especially in sections 2.2 and 2.3.
In section 2.4.1, no details about the concentration of the samples is given.
In section 2.4.5, it is not mentioned how the stability of the micelles have been evaluated.
Also an English revision of this section is needed.
- Other aspects:
The chemical structure of the SF and the surfactants used should be included in the manuscript for clarity.
In line 65, HPMC should be described.
In line 231, The value of 195.67 ± 1.69 does not fits with values in table S1 and Figure S2.
In figure S3, the UV spectra of SF and TPGs is shown. What happens with the UV spectra of the micelles?
In table S1, no information about the units in which size, zeta potential, Dl or EE is shown.
Author Response
Dear reviewer:
We thank you very much for giving us an opportunity to revise our manuscript, we appreciate editor and reviewers very much for their positive and constructive comments and suggestions on our manuscript entitled "Sorafenib nanomicelles effectively shrink tumors by vaginal administration for preoperative chemotherapy of cervical cancer" (ID: nanomaterials-1451585)
we have studied reviewer's comments carefully and have made revision which marked in yellow in the paper. We have tried our best to revise our manuscript according to the comments. Attached please find the revised version, which we would like to submit for your kind consideration.
We would like to express our great appreciation to you and reviewers for comments on our paper. Looking forward to hearing from you.
Thank you and best regards
yours sincerely,
Jun Wang
Corresponding author
Name: Zhe-peng Liu
E-mail: zhepengliu@126.com

Reviewer 2 Report
The clearance to conduct anticancer study in animal models is necessary and the details of it should be reported in the methods section. Also, add the precausions used to conduct study and facilities (type of lab, etc) under which it is performed.
overall, the ms could be considered after minor revision
Author Response

(The authors gave the same response as above.)

Reviewer 3 Report
The manuscript is of interest but some revisions are needed.
Please define VEGF in line 45
Please indicate the type of HPMC used. The unit for viscosity is mPa s and not mpas. Is 4000 mPa s the viscosity of 1% HPMC? Please define.
Line 127-129 The method reported should refer to DSC analyses but it is not specified. Which instrement was employed? Were samples analysed in solid form (for instance after freeze-drying) or in a liquid form?
How was evaluated the stability of systems? Please indicate in the method.
For the release study how much sample was withdrawn at each time point? The medium were replaced? Or not?
Line 225-227 "The beam...solubility" It is not clear the relationship between micelle solubility and light. Micelle are not soluble in water, they are disperse systems.
Line 232 "mass" instead of "mess"
Line 233-234 It is not clear why the entrapment efficiency should decrease, with the larger amount of drug used for the encapsulation experiments, while drug loading increases. Please explain better in the text this aspect.
Line 243-244 it is not clear the description of the spectra reported in Figure S3 in relation to the measurement of sorafenib. Please repharse.
Line 255 and Figure 1E Please explain what refers the transition around 60 °C related to SF micelles.
Line 259-260 Please describe better stability results in PBS and RPMI1640+FBS and FBS.
Line 326-329 Method used for the determination of G' and G'' is not reported in the method section.
Line 329-331 and Figure S4 How was determined the particle size of micelles in HPMC? DLS is a not suitable technique for particle in an hydrogels since they don't move by brownian motion. Typo mistake in Figure S4.
Line 379-381 it is not clear the role of nitrogen in the formation of micelles. Please explain and add reference if already used.
I suggest implementing the discussion section and separate from it the conclusion.
Author Response

(The authors gave the same response as above.)

Round 2
Reviewer 1 Report
The reviewer recommends its publication since the authors have well addressed all the concerns.
Author Response
Dear reviewer:
Thanks very much for your kind work and consideration on publiation of our paper. On behalf of my co-authors, we would like to express our great appreciation to editor and reviewers.
Thank you and best regards
yours sincerely,
Jun Wang
Corresponding author
Name: Zhe-peng Liu
E-mail: zhepengliu@126.com
Reviewer 3 Report
Line 209 and 210 "and tested for elasticity (G') and viscous modulus (G'') by rheology" Which rheological test was performed? At which conditions? G' is the "elastic modulus" and not " elasticity modulus"
Lne 302 Maybe it is "crystallinity" instead of "crystalline"
Typo mistake in Figure S4 was not corrected. The title of Y-axis is "Intensity" and not "Indensity"
The authors have separated discussions from conclusions but they did not implemented the discussion section: Line 453-459 This part of the discussion must be rephrased since it is not clear and, in general, the discussion about the in vivo studies should be implemented.
Author Response
Dear reviewer:
We thank you very much for giving us an opportunity again to revise our manuscript, we appreciate editor and reviewers very much for their positive and constructive comments and suggestions on our manuscript entitled "Sorafenib nanomicelles effectively shrink tumors by vaginal administration for preoperative chemotherapy of cervical cancer" (ID: nanomaterials-1451585)
We have studied reviewer's comments carefully and have made revision which marked in blue in the paper. We have tried our best to revise our manuscript according to the comments. Attached please find the revised version, which we would like to submit for your kind consideration.
We would like to express our great appreciation to you and reviewers for comments on our paper. Looking forward to hearing from you.
Thank you and best regards
yours sincerely,
Jun Wang
Corresponding author
Name: Zhe-peng Liu
E-mail: zhepengliu@126.com

This manuscript is a resubmission of an earlier submission. The following is a list of the peer review reports and author responses from that submission.